

# Importance of vertical mixing and barrier layer variation on seasonal mixed layer heat balance in the Bay of Bengal

Ullala Pathiranage Gayan Pathirana[1, 2], Gengxin Chen[1], Tilak Priyadarshana[2], Dongxiao Wang[1]

[1]State Key Laboratory of Tropical Oceanography, South China Sea Institute of Oceanology, Chinese Academy of Sciences, Guangzhou, China.

[2]Faculty of Fisheries and Marine Sciences & Technology, University of Ruhuna, Matara, Sri Lanka.

*Correspondence to*: Dongxiao Wang (dxwang@scsio.ac.cn)



**Abstract**

Time series measurements from the Research Moored Array for African-Asian-Australian Monsoon Analysis
and Prediction (RAMA) moorings at 15° N, 90° E; 12° N, 90° E; 8° N, 90° E; 4° N, 90° E; 1.5° N, 90° E; 0° N, 90°
E are used to investigate the seasonal mixed-layer heat balance and the importance of barrier layer thickness (BLT)
and vertical mixing ($Q_{-h}$) in the Bay of Bengal (BoB). It is found that the BLT, $Q_{-h}$ and mixed-layer heat balance all
have a strong seasonality in the central BoB. Sea surface temperature (SST), salinity and wind are important for the
observed strongest seasonal cycle of BLT in the central BoB, and wind is more important than the SST in the southern
BoB. The heat storage rate (HSR) is primarily driven by latent heat flux and shortwave radiation ($Q_{SW}$ $and$ $Q_L$).
Seasonal variations and the magnitudes of longwave radiation ($Q_{LW}$), sensible heat flux ($Q_S$), and horizontal mixed-
layer heat advection are much weaker compared to those of $Q_{SW}$ $and$ $Q_L$. $Q_{-h}$ follows a pronounced seasonal cycle
in the central BoB and is significantly positively correlated with the seasonal cycle of BLT at each mooring location.
The seasonal variability of the stability favors the $Q_{-h}$ during winter and summer monsoon and suppress $Q_{-h}$ during
monsoon transition periods. We found that $Q_{-h}$ plays the secondary role in the seasonal mixed-layer heat balance in
the BoB. It is evident from the analysis that $Q_{-h}$ associated with temperature inversion ($\Delta T$) warms the mixed layer
during winter and cools the mixed layer during summer. The warming tendency during winter is strong in the central
BoB and weakens towards the equator, indicating a cooling tendency around the year. Our analysis further indicates
the weakening of $Q_{-h}$ during monsoon transition periods favors the existence of warmer SST in the BoB, associated
with thermal and salinity stratification in the central BoB.



## 1. Introduction

The Bay of Bengal (BoB) is a semi-enclosed basin with unique characteristics due to the influences of Asian Monsoon
and freshwater influx. It is distinguished by a strongly stratified surface layer and seasonally reversing circulation
(Shetye et al., 1996; Schott and McCreary, 2001), and also forced remotely by seasonal winds in the equatorial Indian
Ocean (McCreary et al., 1993). Remote equatorial Kelvin waves influence the BoB via direct contact along the eastern
boundary of the bay (Yu, 2003). Also, the BoB receives a large quantity of fresh water via precipitation and river
runoff which exceed evaporation (Harenduprakash and Mitra, 1988), which makes thesurface layer buoyant and
maintains strong stratification in the upper BoB (Shetye et al., 1996; Agarwal et al., 2012). This strong stratification
maintains the stability in the surface layer (Chowdary et al., 2016) and supports the formation of a barrier layer (BL),
a unique layer between the base of the mixed layer and the top of the isothermal layer (Lukas and Lindstrom, 1991;
Sprintall and Tomczak, 1992; Girishkumar et al., 2013). The presence of a BL restricts the mixing within the mixed
layer, and also affects sea surface temperature (SST) by reducing the mixing of cool thermocline water in the mixed
layer (Vialard and Delecluse, 1998b; Foltz and McPhaden, 2009), hence playing an important role in surface mixed-
layer heat balance (Lukas and Lindstrom, 1991). Using a model simulation, Montegut et al., (2007b) suggested that
thicker BLs are linked to positive SST anomalies in the northern Indian Ocean. Thus, understanding BL formation
and its variabiltiy is important to explain the energy balance in the upper layer of the BoB.

Formation and variability of BL depend on the variability of isothermal layer depth (ILD), which is related to
shoaling thermocline, Ekman pumping (Thadathil et al., 2008), mixed layer depth (MLD) (Vinayachandran et al.,
2002), and wave propagation in the BoB (Chacko et al., 2012). Wind stress acts against the formation of a thick BL
(Bosc et al., 2009) by deepening the mixed layer. Freshwater flux facilitates a thick BL (Cronin and McPhaden, 2002)
by reducing MLD through stratification at the surface. Thus, the seasonality of BL thickness (BLT), which can
influence vertical mixing (Wang et al., 2011), is an important phenomenon in the tropical oceans' surface-layer energy
balance.

Seasonal variations in the parameters controlling the mixed-layer heat balance are associated with changes in the
monsoonal winds over the BoB. McPhaden and Foltz (2013) suggested that in the tropics radiative fluxes dominate
the surface heat flux variation; and Chacko et al., (2012) pointed out the significance of atmospheric forcing which
influence the mixed-layer temperature/SST in the central BoB. Further, Chacko et al., (2012) suggested the importance
of wind-induced heat loss and vertical entrainment in the mixed-layer heat balance. The net radiation increases from
a minimum in winter to a maximum in April-May, and thereafter decreases sharply with the start of summer monsoon.
Generally, during pre-summer monsoon the SST over the BoB is higher than 29°C (Chacko et al., 2012). The surplus
energy at the sea surface warms the surface layer with shoaling ILD. Heat loss during winter cools the surface layer
associated with a thicker BL in the BoB. Using the Research Moored Array for African-Asian-Australian Monsoon
Analysis and Prediction (RAMA) mooring data in the southern BoB (8° N, 90° E), Girishkumar et al. (2013) suggested
the significance of BL and temperature inversion on mixed-layer heat budget. Using the Pilot Research Moored Array
in the Tropical Atlantic (PIRATA) mooring data at three locations (15°N, 38°W; 12°N, 38°W; 8°N, 38°W), Foltz and



McPhaden (2009) pointed out that seasonal variation of BLT can exert a considerable influence on SST through its modulation of the vertical heat flux at the base of the mixed layer in the tropical Atlantic Ocean. The importance of

vertical mixing motivated us to focus on the influence of BLT and vertical mixing on seasonal mixed-layer heat balance in the BoB.

The objectives of this study are to examine the seasonal mixed-layer heat balance and to explain the role of BLT variability and vertical mixing (entrainment + vertical diffusion) using the data from the RAMA moorings at 15°N, 90°E; 12°N, 90°E; 8°N, 90°E; 4°N, 90°E; 1.5°N, 90°E; 0°N, 90°E. We use the measurements from the RAMA

(McPhaden et al., 2009) mooring in the central BoB together with the World Ocean Atlas 2013 (WOA13), TropFlux, QuikSCAT winds, GPCP precipitation, OI SST and NCOM datasets to investigate the seasonal mixed-layer heat balance and the importance of BLT and vertical mixing in the BoB. In section 2, data and methods used in the analysis are described. Results are presented in section 3. We provide a summary in section 4, which is followed by conclusions.

## 2.   Data and Methods

Data from multiple sources are utilized in this study. Daily time series data from 01 January 2008 to 01 January 2017 obtained from the RAMA moorings deployed along 90°E, from the central BoB to the equator (Figure 1) are used to estimate daily MLD, ILD, BLT, and the terms in mixed-layer heat balance. Measurements include SST, subsurface temperature and salinity, air temperature, sea level pressure, wind velocity, relative humidity, shortwave and longwave radiation, 20°C isothermal layer, and surface currents. Ocean currents are collected at 10-m depth.

Meteorological measurements are collected 3-4 m above the sea surface. Ocean temperatures are measured at 1, 5, 10, 13, 20, 40, 43, 60, 80, 100, 120, 140, 180, 300, and 500 m. Salinity data is collected at 1, 5, 10, 20, 40, 60, 100, and 140 m. Missing values are replaced with values calculated from least square approach and interpolation. The data are linearly interpolated vertically to 1-m interval. All the datasets are filtered using 3-month running mean (Foltz and

McPhaden, 2009) before further calculations to identify seasonal cycles. Temperature and salinity profiles from the WOA13 dataset (from 2005 to 2012) (Locarnini et al., 2013) are used to obtain monthly climatologies of MLD, ILD and BLT in the BoB.

MLD is defined as the depth where the density has changed by 0.125 kg m$^{-3}$ ($\Delta T = 0.8\ ^\circ C$) from the surface value at 1 m (Rao and Sivakumar, 2000; Kara et al., 2003). This criterion consider both temperature and salinity effects on

stratification. MLD is estimated using Eq. (1).

$$\Delta \sigma_t = \sigma_t\ (T + \Delta T, S, P) - \sigma_t\ (T, S, P) \qquad (1)$$

where $S$ is salinity, $T$ is temperature and $P$ is pressure. ILD is calculated as the depth where temperature is 0.8°C lower than the SST ($\Delta T = 0.8°C$) (Du et al., 2005). BLT is defined as the difference of ILD and MLD. $BLT = ILD - MLD$ (Sprintall and Tomczak, 1992; Girishkumar et al., 2013). Chacko et al. (2012) poninted out that the estimation of MLD

and ILD by 0.8°C criterion is reasonably good for the BoB compared to 0.5°C or 1.0°C criterion.



Air-sea fluxes at the mooring locations are computed from the daily winds extrapolated to the 10-m height, SST, air temperature and relative humidity. Latent ($Q_L$) and sensible ($Q_S$) heat fluxes are estimated using the Coupled Ocean-Atmosphere Response Experiment (COARE) bulk algorithms (Fairall et al., 2003). As only the mooring at (15°N, 90°E) had longwave radiation ($Q_{LW}$) measurements, net longwave radiation from the TropFlux is used (Kumar et al., 2011) for other mooring locations. Net shortwave radiation ($Q_{SW}$) is estimated from downwelling shortwave radiation measured at the mooring sites, corrected for albedo (6%) at the surface. The heat flux into the mixed layer due to air-sea exchange ($Q_{net}$) is estimated using the equation (Eq. 2) below,

$$Q_{net} = Q_{SW} + Q_{LW} + Q_L + Q_S \qquad (2)$$

With the convention that heat flux is positive when it is into the ocean. The penetrating shortwave radiation below the mixed layer is estimated using $Q_{pen} = 0.47 \times Q_{sw} . e^{-kh}$ (Jouanno et al., 2011), considering a constant $e^-$ folding depth of 25 m ($k=0.04$) and h (MLD).

To address the seasonal variability of the mixed-layer heat balance at each mooring location, we consider the following expression (Rao and Sivakumar, 2000; Foltz and McPhaden, 2009; Girishkumar et al., 2013; McPhaden and Foltz, 2013),

$$\rho C_P h \, \frac{\partial T}{\partial t} = Q_0 - \rho C_P h \left[ u \frac{\partial T}{\partial x} + v \frac{\partial T}{\partial y} \right] + q_{-h} + \in \qquad (3)$$

The terms in (Eq. 3) represent, from left to right, heat storage rate (HSR), net surface heat flux, horizontal mixed-layer heat advection, and the combination of entrainment and vertical turbulent heat fluxes at the base of the mixed layer. The error estimation ($\in$) includes errors in the estimation of the terms in (Eq. 3), which are associated with data sources and unrepresented/unresolved physical processes (Foltz and McPhaden, 2009). In (Eq. 3), $T$ is averaged mixed-layer temperature, $\rho$ is density of seawater ($\rho = 1024 \, kg \, m^{-3}$), $C_p$ is specific heat capacity of seawater ($C_p = 4000 \, J \, kg^{-1} \, K^{-1}$), $h$ is MLD, $t$ is time, and $Q_0$ is the surface heat flux adjusted for the penetrative shortwave radiation through the base of the mixed layer. Heat fluxes at the mooring locations are estimated using the terms in (Eq. 2). The zonal ($u$) and meridional ($v$) component of current measurements at 10-m depth are obtained from the moorings. Assuming that the temperature is uniformly distributed in the entire mixed layer, we use Optimally-interpolated SST (OI SST) product with 0.25° × 0.25° resolution (Figure 2) to compute the horizontal advection term in Eq. (3). OI SSTs averaged over 50 km on either side of the mooring locations are used to estimate the horizontal gradient of SST (Vialard et al., 2008; Girishkumar et al., 2013).

Further we use the Navy Coastal Ocean Model (NCOM) monthly climatological data (from 1990 to 2011) (Ke Huang et al., 2015) to compare with the observed seasonal variability in upper layer stratification, subsurface temperature and salinity gradients, and the stability at 15°N, 90°E by the RAMA mooring. The water stability (E) in terms of the buoyancy frequency is estimated using (Murty et al., 1996),

$$E = \frac{g}{\rho *} \left[ \frac{d\rho}{dz} - \frac{g\rho^*}{c^2} \right] \qquad (4)$$





The terms in Eq. (4) represent, $g$ is the acceleration due to gravity (9.8 ms$^{-2}$), $\rho$ is the sea water density (kgm$^{-3}$), $\rho *$ is the mean density in the water column $dz$, $z$ is the depth (m), and $C$ is the sound velocity (ms$^{-1}$) in the sea

water (Chen and Millero, 1977). The Quick Scatterometer (QuikSCAT) winds, Global Precipitation Climatology Project (GPCP) precipitation, Optimum Interpolation Sea Surface Temperature (OI SST), SSS, and BLT climatologies are used to describe the basin-scale seasonal cycle in the BoB (Figure 1, 2).

## 3. Results

### 3.1. Variability of climatological surface conditions in the BoB

Before examining the variability of conditions at the RAMA mooring locations, we examine the climatological seasonal conditions in the BoB. Figure 1 shows the seasonal climatologies of BLT, precipitation and winds in the BoB. Figure 2 illustrates the seasonal climatology of salinity and SST anomalies in the bay. During winter monsoon (December-February), relatively larger BLT is present from the central to northern BoB (Figure 1a), when

the SST cooling is the largest (Figure 2a) and there is relatively low precipitation. Seasonal cycle of salinity shows freshening in the northern BoB during winter (Figure 2a), which indicates the importance of river runoff associated with surface cooling for the formation of thicker BL in the BoB. BL almost disappears during pre-summer monsoon (March-April) associated with low precipitation (Figure 1b), low surface freshening and warmer SST (Figure 2b). During summer monsoon (May-September), BLT is relatively thicker in the eastern boundary of the BoB, associated

with higher precipitation (Figure 1c) and surface freshening (Figure 2c). During post-summer monsoon (October-November), BLT varies between 0-30 m (Figure 1d), associated with low salinity in the northern BoB and with SST cooling in the southern BoB (Figure 2d). Seasonal climatology of wind clearly indicates the weakening or strengthening of surface winds over the BoB (Figure 1), which is one of the important factors for the formation of thicker or thinner BL (Bosc et al., 2009).

Next, we examine the stratification in the upper 140 m at the RAMA locations in the BoB. The moorings are located from the central BoB to the equator, which are in a region of strong seasonal variability of BLT (Figure 1). We have selected the mooring at 15°N, 90°E (Figures 3a, 3b), which has the longest data availability and the significant seasonal cycle of BLT and the mooring at 12°N, 90°E (Figures 3c, 3d) to illustrate the seasonal stratification in the central BoB. The estimated mean errors in MLD and ILD are typically ±2 m and ±3 m with a standard deviation

of ±8 m and ±18 m, giving errors in BLT of around ±5 m with a standard deviation of ±19 m. MLD in the central BoB exhibits a prominent seasonal variation (Figure 6d) with surface freshening and wind forcing, reaching a maximum in July (42±8 m) when the wind is at its maximum (Babu et al., 2004). ILD varies out of phase with SST, reaching its maximum (87±18 m) in February when SST is minimum and reaching its minimum (16±18 m) in April when SST is at its maximum (Figure 6). The seasonal cycle of BLT varies with ILD, reaching its maximum (69±19 m) in February

(highest ILD) and its minimum (2±19 m) in April (lowest ILD) (Figure 3b) then with MLD during summer. Towards the southern BoB, the variability of BLT varies in phase with MLD. Thus, it indicates that SST, salinity and wind are important for the observed strongest seasonal cycle of BLT in the central BoB, and that wind is more important than SST for the variability of BLT in the southern BoB (Girishkumar et al., 2011; Felton et al., 2014).



Then we use the NCOM model estimations to compare the seasonal variability of the conditions observed at

15°N, 90°E in the BoB (Figure 4). Monthly climatology of MLD is over estimated in NCOM compared to that of

RAMA, where the difference is larger during winter (Figure 4a). The estimated ILD agrees well indicating that the

effect from temperature is similar in both data sources (Figure 4b), hence it is evident that the difference observed in

the MLD estimations are associated with the effect due to salinity. NCOM under estimate the BLT during post-summer

monsoon and winter, which indicates the effect of MLD estimation (Figure 4c). Though there are differences in the

magnitudes, the observed seasonal variability in the upper layer stratification at 15°N, 90°E by the RAMA mooring

is evident from the NCOM model estimations. The computed vertical temperature gradient ($dT/dz$) (Figure 5a) and

salinity gradient ($dS/dz$) (Figure 5b) illustrate the seasonal variability of sub surface conditions. It is evident from both

data sources, the existence of homogenous layer during summer which coincide with maximum wind speed and the

highest MLD. The thermal stratification is largest during pre-summer monsoon (Figure 5a) which coincides with the

time period of higher insolation and the salinity stratification is largest during post-summer monsoon (Figure 5b). The

estimated upper ocean stability illustrates that the upper ocean layers at 15°N, 90°E are more stable during monsoon

transition periods compared to that of winter and summer (Figure 5c). Thus the results pointed out that winter and

summer favors the vertical mixing (Thangaprakash et al., 2016) with the presence of more unstable layers in the central

BoB, and pre and post-summer monsoon tends to inhibit the vertical mixing due to the presence of more stable water

layers.

### 3.2. Mixed-layer heat balance

Measurements from 15°N, 90°E and 12°N, 90°E reveal pronounced seasonal cycles of SST, wind speed, net

surface heat flux, MLD, and ILD during 2008-2016 (Figure 6). SST reaches its maximum (30.4±1°C) in pre-summer

monsoon and its minimum (26.6±1°C) in winter (Figure 6a). Wind speed reaches its maximum (9.2±1.8 ms$^{-1}$) during

summer and its minimum (2.9±1.8 ms$^{-1}$) in pre-summer monsoon (Figure 6b). The surface heat flux follows the

seasonal cycle of SST, tends to heat the mixed layer during the pre-summer through post-summer monsoon and tends

to cool the mixed layer during winter (Figure 6c). Both MLD and ILD vary out of phase with wind and SST, where

the seasonal cycle is strong in the central BoB (Figure 6d). Higher differences in SST (~1°C) are observed during

winter, from the central to southern BoB; and less differences in SST (<0.5°C), during pre- to post-summer monsoon.

The mean SST at the mooring locations increases towards the equator, and the moorings located at 15°N and 12°N

experience the highest SST during pre-summer monsoon. Wind speed undergoes a more pronounced seasonal cycle

at 15°N and 12°N compared to that at 8°N, 4°N, 1.5°N and 0°N, tending to enhance the seasonal cycle of $Q_L$ in the

central BoB.

Based on the strongest seasonal cycles observed, we consider the mixed-layer heat balance at 15°N, 90°E

(Figure 7). The seasonal cycle of SST at 15°N, 90°E is driven primarily by changes in the net surface heat flux.

McPhaden and Foltz [2013] suggested that in the tropics radiative fluxes ($Q_{SW}$ and $Q_{LW}$) dominate the surface heat

flux variation. Net surface shortwave radiation is the strongest (271±34 W m$^{-2}$) in pre-summer monsoon, and the

amount of shortwave radiation absorbed (196±21 W m$^{-2}$) by the mixed layer also peaks during this season (Figure 7a),

coinciding with a minimum in BLT (2±19 m) and low MLD (11±9 m). $Q_L$ tends to cool the mixed layer, and its



magnitude is the largest (-175±29 W m$^{-2}$) during winter (Figure 7b) when the northeasterly wind is strong over the bay. Seasonal variations and magnitudes of $Q_{LW}$ (-30 to -81 ± 81 W m$^{-2}$), $Q_S$ (-3 to -12 ± 2 W m$^{-2}$), and horizontal mixed-layer heat advection (3 to -6 ± 2 W m$^{-2}$) are much weaker (Figures 7b, 7c) compared to those of $Q_{SW}$ $and$ $Q_L$. Thus, these results indicate that the pronounced seasonal cycle of mixed-layer heat storage rate is driven primarily by the seasonal variability of $Q_{SW}$ $and$ $Q_L$ in the central BoB (Figure 7d). Mixed-layer heat storage rate (HSR) reaches

its maximum (43±24 W m$^{-2}$) in pre-summer monsoon and its minimum (- 42±24 W m$^{-2}$) in summer (Figure 7d). Thangaprakash et al. (2016) suggested that the penetrative component of shortwave radiation ($Q_{pen}$) plays a crucial role in the mixed-layer heat balance in the BoB, especially during the pre-summer monsoon and it is evident from our results (Figure 7a). During winter, HSR illustrates a cooling tendency in the central BoB (Figure 7d), but its magnitude is still larger than that of the net surface heat flux. Though HSR is primarily driven by $Q_{SW}$ $and$ $Q_L$, the difference

observed during summer and winter monsoons brings the importance played by other terms in mixed-layer heat balance as secondary.

         The observed mixed-layer heat advection at the mooring locations is much weaker (Figure 8), and the contribution to the mixed-layer heat balance is less significant, illustrating the importance of vertical mixing. To study the importance of entrainment and vertical turbulent heat fluxes (hereafter vertical process) at the base of the mixed

layer, we compute the vertical process following the methods used in Foltz and McPhaden (2009) and Girishkumar et al. (2013). The estimated $Q_{-h}$ (Figure 8) following Foltz and McPhaden (2009) is the difference between the mixed-layer heat storage rate and the sum of the first two terms in equation (3). The estimated vertical process following Girishkumar et al. (2013) is the summation of entrainment and vertical diffusion ($H\left[W_h + \frac{dh}{dt}\right]\frac{(T-T_h)}{h} + \frac{K_z}{h}\frac{\partial T}{\partial z} + \in$). The residual ($\in$) term is larger  compared to the calculated entrainment and vertical diffusion at the mooring locations,

showing the uncertinities due to unpresented/unresolved physical processes in the mixed-layer heat balnce.

         The $Q_{-h}$ term undergoes a strong seasonal cycle at 15°N, 90°E, tending to cool the mixed layer at a rate up to 66±38 Wm$^{-2}$ during summer and to warm the mixed layer up to 70±38 Wm$^{-2}$ during winter (Figure 8a). Contribution by the vertical process to the mixed-layer heat balance decreases towards the equator (Figure 8), indicating the dominance of atmospheric forcing in influencing the mixed-layer heat balance. BLT varies in phase with $Q_{-h}$, reaching

its maximum of 30–70 m during winter and its minimum of <3 m in pre-summer monsoon (Figures 3b, 7a). The correlation coefficient for daily-averaged BLT and $Q_{-h}$ is 0.84 (Table 1). Foltz and McPhaden (2009) suggested the warming and cooling tendencies by $Q_{-h}$ are associated with the temperature differences ($\Delta T$) at the base of the mixed layer and the BLT can exert a significant influence on $Q_{-h}$ through its modulation of $\Delta T$. Thus, it indicates the importance of $\Delta T$ (both positive and negative) associated with the variability of BLT to the mixed-layer heat balance

in terms of $Q_{-h}$ (Figure 9a).

         The combination of surface cooling and a thicker BL is important for the generation of temperature inversion at the base of the mixed layer. During winter, temperature inversion is prominent (Montegut et al., 2007b; Girishkumar et al., 2011; Girishkumar et al., 2013) in the region. Girishkumar et al. (2013) pointed out during times when thicker BL and temperature inversion ($\Delta T$) occur coincidentally, the vertical process shows a strong warming tendency due



to vertical mixing of warm subsurface water into the mixed layer. Montegut et al. (2007b) suggested that temperature inversion supported by BL in the BoB affects surface temperature through entrainment of warm subsurface waters into the mixed layer. We observed the warming tendencies by the vertical process during winter at 15°N, 12°N and 8°N in the BoB similar to that of Foltz and McPhaden (2009) for the tropical Atlantic and of Vialard and Delecluse (1998b) for the western tropical Pacific. Further our results indicate that the vertical process tends to cool the mixed

layer at 4°N, 1.5°N and 0°N around the year. During summer, vertical mixing tends to cool the mixed layer in the central BoB (Figure 8a), reaching its mximum during June. The rate of cooling by vertical mixing in summer decreases from the central BoB to equatorward. Pre-summer through summer monsoon is a period with high net heat flux, tending to increse the temperature in the mixed layer in the central BoB. Thus, it points out vertical mixing during summer plays the secondary role in mixed-layer heat balance.

The effect from the vertical process to the mixed-layer heat balance during monsoon transition periods remains relatively small compared to the other seasons, while during June vertical pocess is more important compared to other terms in mixed-layer heat balance (Figure 9c). The role of the vertical processes during monsoon transition periods is important due to the presence of persistent warm SST in the BoB, higher than 28°C (Shenoi et al., 2002), which is generally considered as the threshold for atmospheric convection (Johnson and Xie, 2010). Figure 9b

illustrates the terms in the vertical process, computed following Girishkumar et al. (2013) and Zeng and Wang (2016), averaged using a 7-day running mean filter to identify the variation. $Q_{-h}$ term calculated following Foltz and McPhaden (2009) indicates the seasonal cycle of the vertical process at 15°N, 90°E. During pre-summer monsoon, $Q_{-h}$ changes its phase from warming (winter) through cooling (summer), indicating the seasonal variability associated with temperature inversion and BLT (Figure 9b). From August to September, the contribution of $Q_{-h}$ indicates a

warming tendency; and during October, there is a cooling tendency with less significance. The contributions from entrainment and vertical diffusion illustrate a cooling tendency during August-September, which indicates a missing source of warming in the central BoB. Foltz and McPhaden (2009) pointed out that the missing sources of warming and cooling found at the PAIRATA moorings in the tropical Atlantic are mainly due to differences in the estimation of horizontal eddy heat advection and penetrative shortwave radiation.

**3.3. Importance of vertical process during post-summer monsoon in the central BoB**

The RAMA moorings at 15°N, 90°E; 12°N, 90°E and 8°N, 90°E illustrate a positve heat storage rate during August–October, which is the period that the central BoB shows the second highest SST due to positive net surface heat flux (Figure 6). Late phase of summer monsoon and post-summer monsoon are an important period for the formation of deep depressions and cyclones in the region. Deepening of the ocean mixed layer associated with SST

cooling is thought to inhibit cyclone intensification (Emanuel, 1999; Wentz et al., 2000). Shay et al. (2000) pointed out that if SST does not cool because the upper ocean has a deep warm water layer, a cyclone could intensify rapidly and sustains its intensity longer. Our results illustrate the shallowest MLD (Figure 6d), associated with a moderate BL (Figure 8), forms during this season in the central BoB. Thus, stratification helps to maintain a warm surface layer, inhibiting SST cooling, and contribution from vertical mixing remains minimum during this period (Figures 8a, 9b).

Chacko et al. (2012) suggested the siginificance of vertical mixing over surface forcing in inducing mixed-layer





temperature variability in the BoB and Sengupta et al. (2008) pointed out that during post-summer monsoon the northern BoB responds quite differently to cyclones than during pre-summer monsoon. Stability during post-summer monsoon due to salinity stratification is stronger compared to that of pre-summer monsoon due to thermal stratification and that may be the reason for the observed differences during cyclone events. Observations from moored RAMA
buoys revealed that the importance of seasonal vertical process in SST cooling/warming associated with BLT in the BoB.

## 4. Summary and Conclusions

In this study, we examine the seasonal mixed-layer heat balance and the importance of the vertical process and BLT in the BoB, using time series measurements recorded at 15°N, 90°E; 12°N, 90°E; 8°N, 90°E; 4°N, 90°E;
1.5°N, 90°E; 0°N, 90°E by the RAMA moorings. At all the mooring locations, it is found that the seasonal changes in mixed-layer HSR is primarily driven by shortwave radiation ($Q_{SW}$) and latent heat flux ($Q_L$). The seasonality of HSR is more pronounced in the central BoB. Seasonal variations and magnitudes of longwave radiation ($Q_{LW}$), sensible heat flux ($Q_S$) are smaller compared to those of $Q_{SW}$ and $Q_L$. The horizontal mixed-layer heat advection also weaker compared to that of vertical mixing. The vertical mixing at the base of the mixed layer ($Q_{-h}$), estimated
as the residual in the heat balance following Foltz and McPhaden (2009), also follows a pronounced seasonal cycle in the central BoB, and is correlated positively with the seasonal cycle of BLT at each mooring location. We find that $Q_{-h}$ plays the secondary role in mixed-layer heat balance in the BoB. It is evident from the analysis that the vertical mixing associated with temperature inversion ($\Delta T$) warms the mixed layer during winter and cools the mixed layer during summer. The warming tendency during winter is strong in the central BoB and weakens towards the equator,
indicating a cooling tendency around the year. The impact of BLT on $Q_{-h}$ is the strongest at 15°N, 90°E where the seasonal cycle of BLT is the strongest, which is consistent with the results of Foltz and McPhaden (2009) in the central tropical Atlantic.

To examine the importance of entrainment and vertical diffusion in the vertical process, we estimated vertical mixing following Girishkumar et al. (2013), and found that entrainment is more important in the vertical process. We
have found a missing source of warming during August–September in the central BoB up to ~25 Wm⁻². The uncertinities are mainly associated with measurement errors, calculation errors and parameterization of the vertical process. Our results further indicate that entrainment is weaker during post-summer monsoon period, which tends to weaken the SST cooling by vertical mixing and helps to maintain warmer surface temperature at all the locations. The seasonal variability of the upper ocean stability favors the $Q_{-h}$ during winter and summer monsoon and suppress $Q_{-h}$
during monsoon transition periods in the BoB. The surface heat fluxes alone do not account for the changes observed in seasonal mixed-layer heat balance. Thus, it brings the importance of vertical mixing, which influences the seasonal variability of mixed-layer heat balance in the BoB.

This study further indicates that MLD, ILD and BLT undergo a strong seasonal cycle in the central BoB. It is evident from our results the change in ILD with SST is important for the change in BLT during winter and pre-
summer monsoons, while the change in MLD with wind and surface freshning is important during summer and post-



summer monsoons in the central BoB. The significant positive correlation between BLT and $Q_{-h}$ means that vertical mixing is the weakest when the BL is the thickest. We have found that, time periods with the thicker and thinner barrier layers are associated with significant vertical mixing where the moderate BLT supresses the vertical mixing in the central BoB during the periods with strong upper ocean stability. The warming and cooling tendencies by vertical

mixing associated with the variability of BLT in the central BoB are consistent with the results of Vialard and Delecluse (1998b) in the western equatorial Pacific and Foltz and McPhaden (2009) in the central tropical Atlantic. Thus, it illustrates the imporatnce of the seasonal cycle of BLT on the mixed-layer heat balnce in the central BoB.

The results of this study thus indicate the importance of BLT and vertical mixing on the seasonal mixed-layer heat balance in the BoB. Late phase of summer monsoon and post-summer monsoon are a period of active air-sea

interaction in the BoB, and it is possible that weakening of vertical mixing and strong stratification (higher stability) during this period influence the intensity and frequency of BoB cyclones. Moreover, studies with systematic measurements are needed to understand the upper-ocean dynamics, the process of vertical mixing and its influence on mixed-layer temperature in the BoB, which can influence the weather and climate in the region and beyond.






**Data availability**

Access to data sources used in this study are available through the links provided in the acknowledgement.

**Competing interests**

None








**Acknowledgement**

This work is supported by the NSFC (grants 41476011 and 41521005), XDA (grant 11010102) and the Youth
Innovation Promotion Association of CAS (2017397). The RAMA data are provided by the TAO Project Office of
NOAA/PMEL (www.pmel.noaa.gov). WOA V2 2013 data are provided by NOAA/NODC (www.nodc.noaa.gov).
The Argo data were collected and made freely available by the International Argo Program and the national programs
that contribute to it (www.argo.ucsd.edu, argo.jcommops.org). The Argo Program is part of the Global Ocean
Observing System. The TropFlux data are produced under a collaboration between LOCEAN from Institut Pierre
Simon Laplace (IPSL, Paris, France) and the National Institute of Oceanography/CSIR (NIO, Goa, India)
(www.incois.gov.in). The OI SST data are downloaded from www.esrl.noaa.gov. The QuikSCAT data, GPCP
precipitation and BLT climatology are obtained from the University of Hawaii (www.apdrc.soest.hawaii.edu).






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






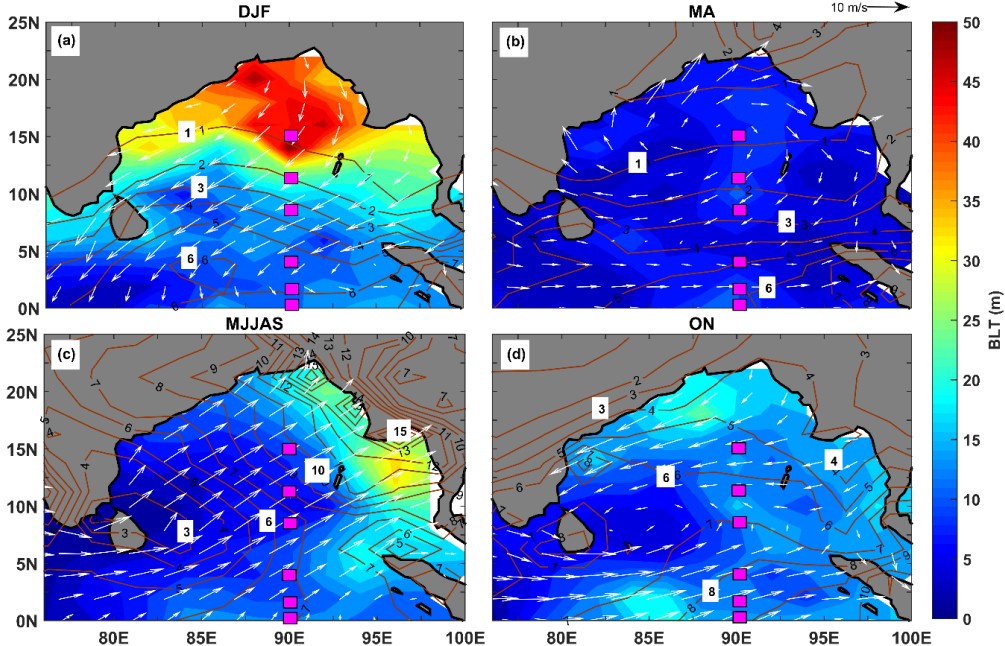

**Figure 1.** Seasonal climatological of BLT (color shaded), precipitation (contour (mm/day)) and wind (arrows) (a) winter monsoon, (b) pre-summer monsoon, (c) summer monsoon, and (d) post-summer monsoon, from Ocean Climatology, GPCP and QSCAT for the BoB. The pink square indicates the locations of the RAMA moorings (15 °N, 90 °E; 12 °N, 90 °E; 8 °N, 90 °E; 4 °N, 90 °E; 1.5 °N, 90 °E; 0 °N, 90 °E) used in this study.








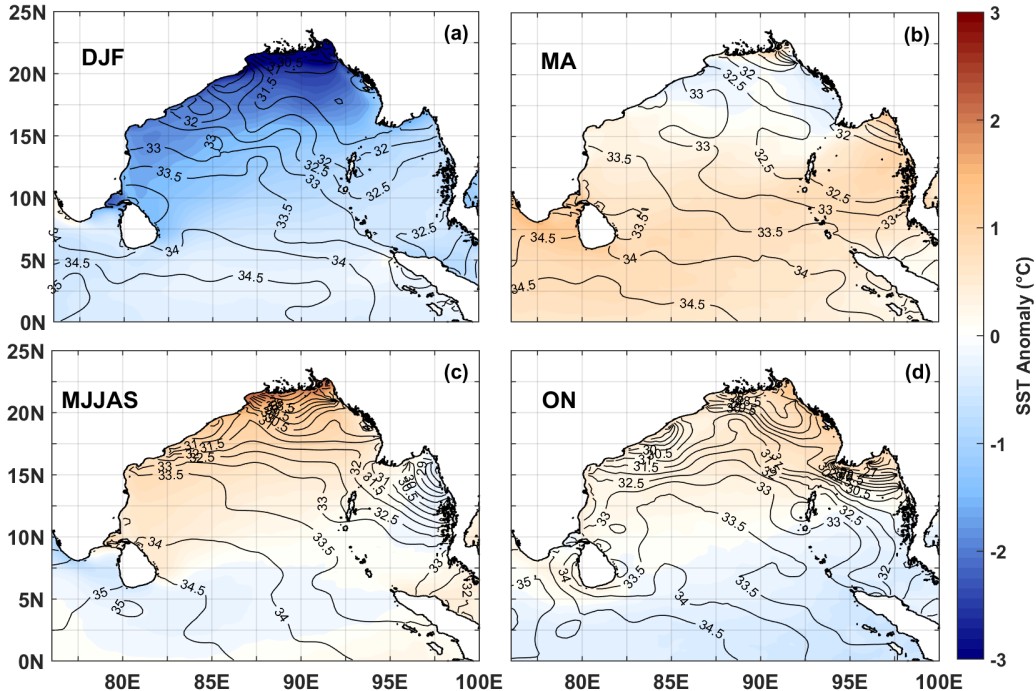

**Figure 2.** Seasonal anomaly of SST (color shaded) and salinity (contour) (a) winter monsoon, (b) pre-summer monsoon, (c) summer monsoon, and (d) post-summer monsoon, from OI SST (2008-2016) and WOA13 (2005-2012) for the BoB.





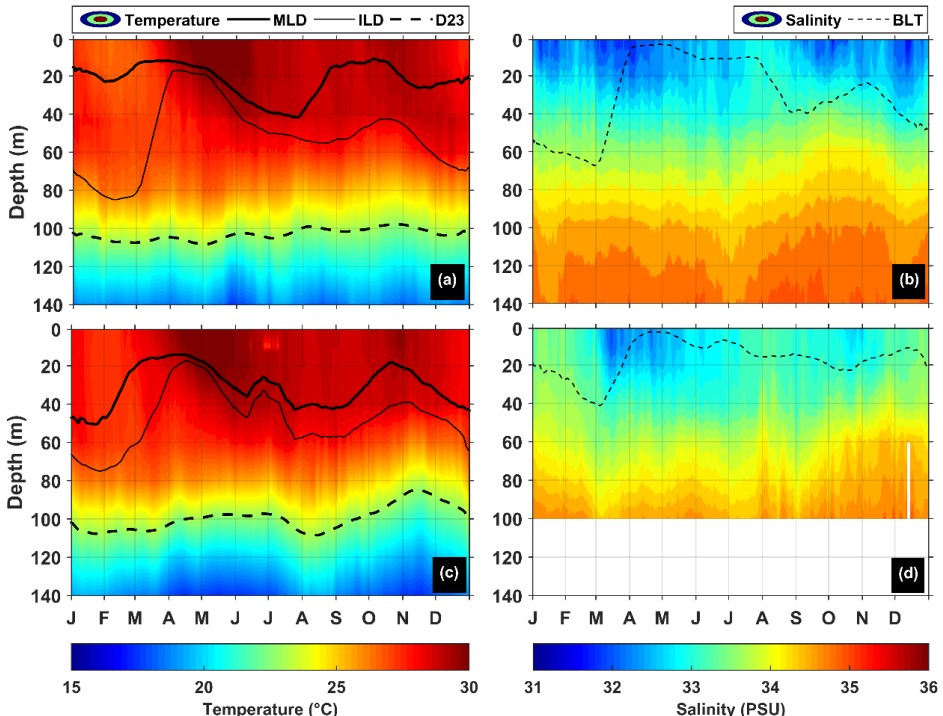


**Figure 3.** Time series of daily averaged data from January 2008 – January 2017 (a, b) daily RAMA buoy data at 15 °N, 90 °E, (c, d) daily RAMA buoy data at 12 °N, 90 °E, (a, c) sub surface temperature, (b, d) sub surface salinity. In the figure (a, c) thick, thin and dashed lines indicate MLD, ILD and D23, (b, d) dashed line indicate BLT.





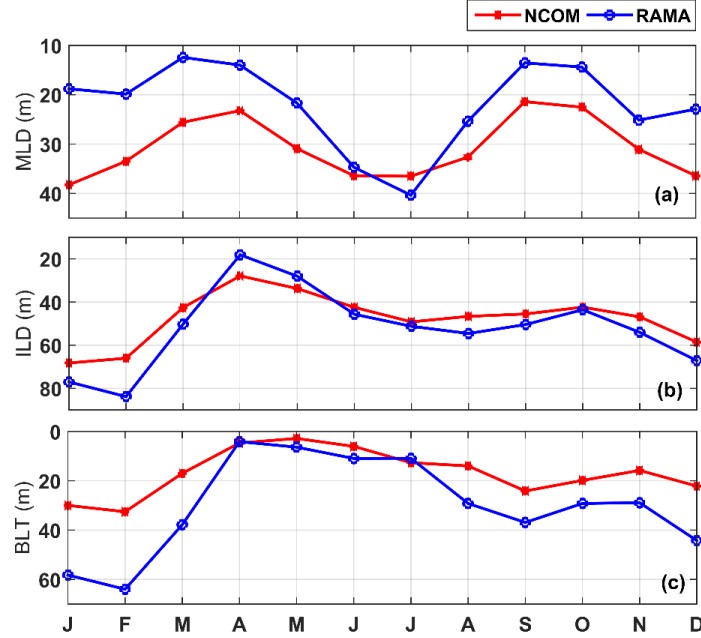

**Figure 4.** Comparison of NCOM (red) and RAMA (blue) estimated monthly climatologies of (a) MLD, (b) ILD, and

(c) BLT.







**Figure 5.** Comparison of upper ocean stability estimated from NCOM (contour) and RAMA (color shaded) at 15°N, 90°E. (a) Temperature gradient (positive when temperature decreases downward), (b) salinity gradient (positive when salinity increases downward), and (c) stability (in terms of buoyancy frequency).





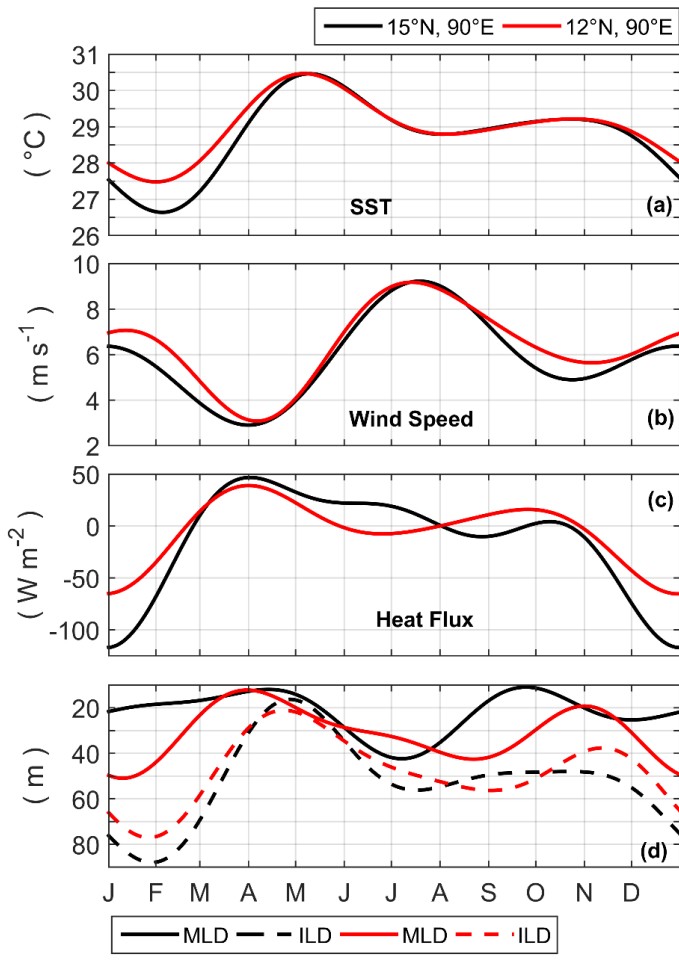

**Figure 6.** Seasonal cycles of (a) SST, (b) wind speed, (c) net surface heat flux, (d) mixed layer depth (**solid**) and isothermal layer depth (**dashed**) from RAMA moorings at 15°N (black), and at 12 °N (red). All-time series have been smoothed using 3-month running mean filter.






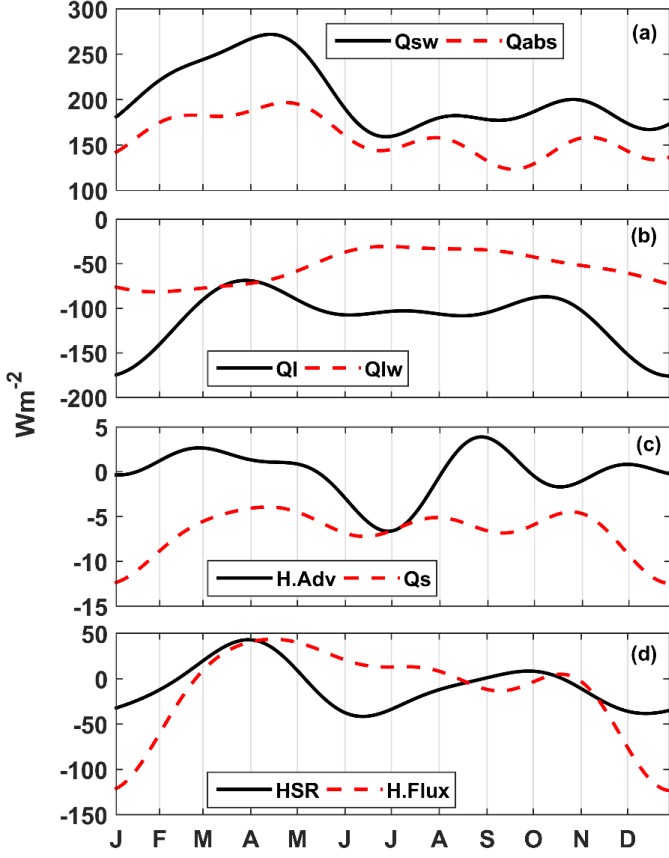

**Figure 7.** Daily averaged time series data from RAMA at 15 °N, 90 °E (a) net shortwave radiation ($Q_{sw}$) and short
wave radiation absorbed in the mixed layer ($Q_{abs}$), (b) latent heat flux ($Q_l$) and net longwave radiation ($Q_{lw}$), (c)
horizontal advection (H.Adv) and sensible heat flux ($Q_s$), (d) heat storage rate (HSR) and net surface heat flux
(H.Flux). Negative values indicate heat loss from the mixed layer. All-time series have been smoothed using 3-month
running mean filter.






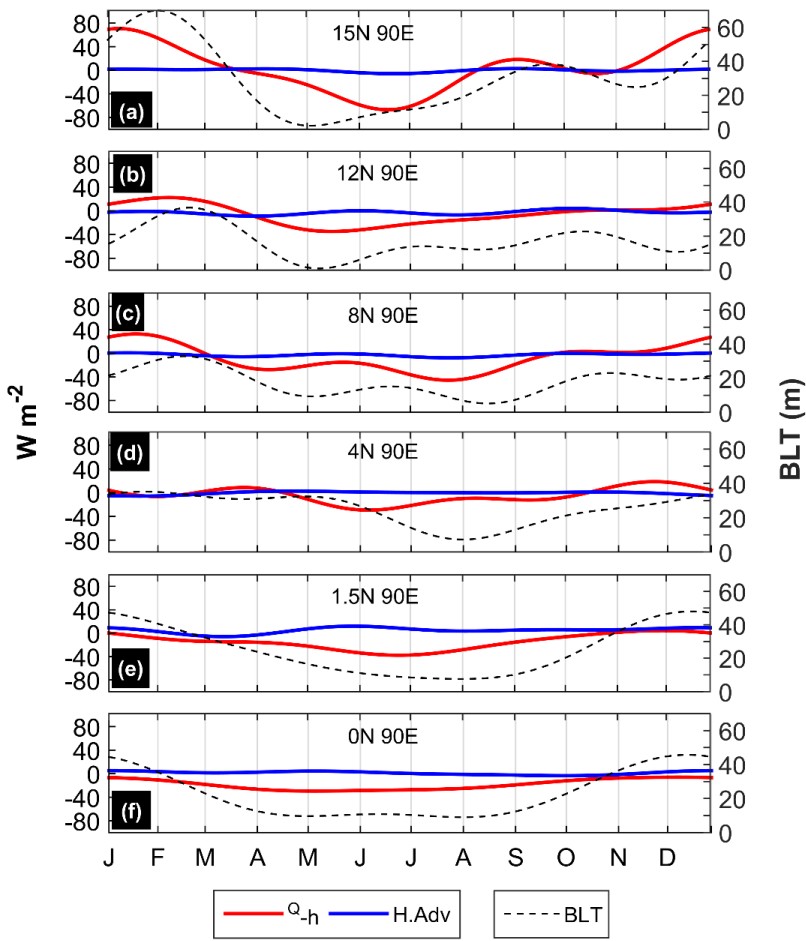

**Figure 8.** Seasonal cycles of vertical process ($Q_{-h}$), horizontal advection (H.Adv) and BLT (a) 15 °N, (b) 12 °N, (c) 8
°N, (d) 4 °N, (e) 1.5 °N, and (f) 0 °N. All-time series have been smoothed using 3-month running mean filter.





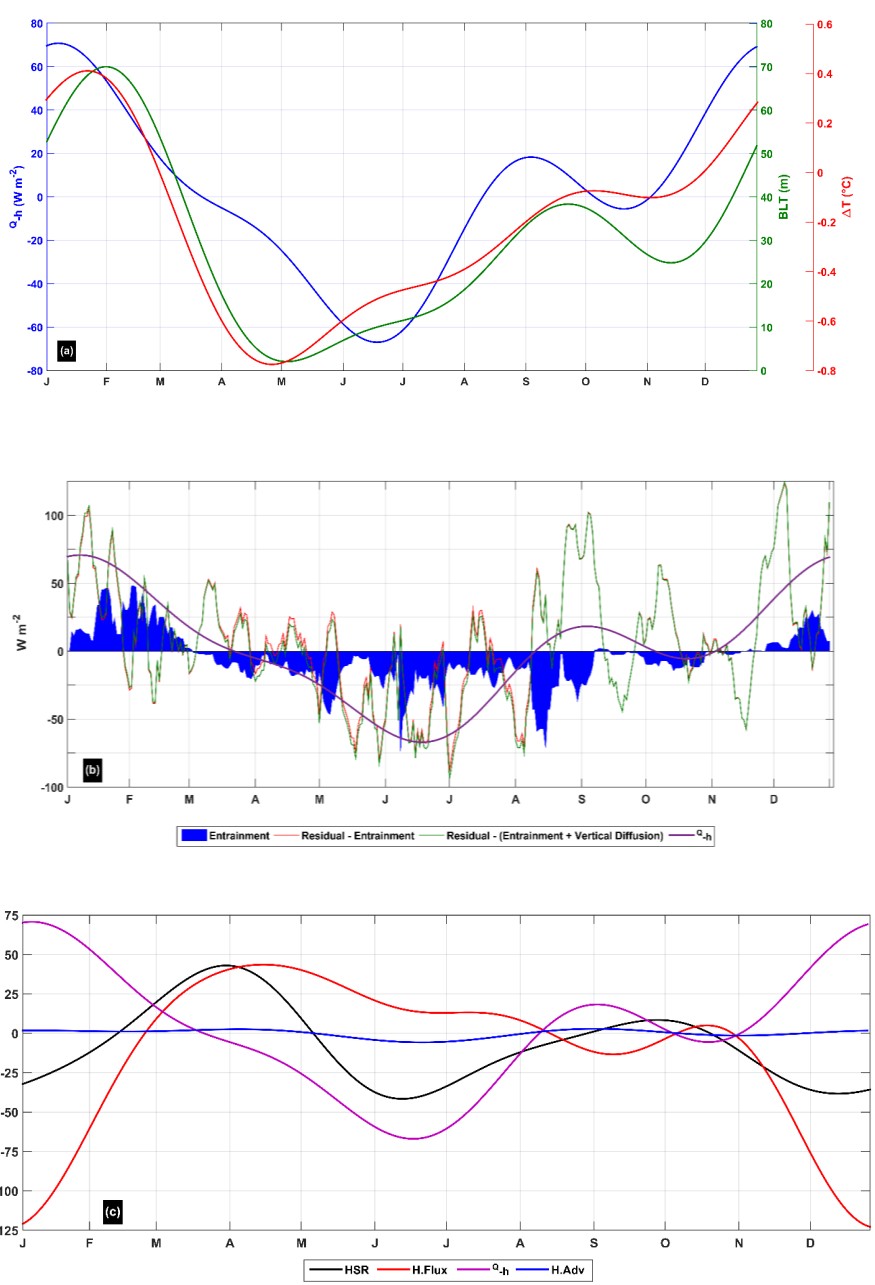

**Figure 9.** Seasonal cycles (a) vertical process ($Q_{-h}$) (blue), temperature difference (red) and BLT (green), (b) entrainment (blue), residual – entrainment (red), residual (R) – entrainment (E) & vertical diffusion (VD) (green), and vertical process ($Q_{-h}$) (purple), (c) mixed-layer heat balance terms at 15°N,90°E. All-time series in (a, c) have been smoothed using 3-month running mean filter. Time series in (b) have been smoothed using 7-day running mean filter.




**Table 1.** Correlation between BLT and $Q_{-h}$ at RAMA mooring locations.

|  | 15 °N, 90°E | 12 °N, 90°E | 8 °N, 90°E | 4 °N, 90°E | 1.5 °N, 90°E | 0 °N, 90°E |
|---|---|---|---|---|---|---|
| **Correlation value (r)** | 0.84 | 0.84 | 0.74 | 0.40 | 0.84 | 0.93 |