# Peer review of "Importance of vertical mixing and barrier layer variation on seasonal mixed layer heat balance in the Bay of Bengal"

_Ocean Science, 2017_

## Referee Comment (RC1) · Anonymous Referee #1 · 25 Sep 2017

This is a very interesting work to assess the mixed layer temperature variations in the Bay of Bengal, emphasizing the role of the barrier layer. I have some minor comments for the authors to consider in revising the manuscript.

It is useful to state clearly what's new in the study and what has been concluded from the previous study in both the Abstract and Summary. I would suggest separating the "Summary and conclusion" section into "Summary" and "Discussion", which will make the reader easy to follow the new findings.

Figure 3: It is useful to show the monthly wind speed in Figure 3 at the selected sites in order to make the statement that the wind speed is the main driver of the barrier layer

[Figure]

thickness variations in line 192, or make the statement to the discussion.

The paragraph starting from line 180: it is not clear how the error bars are calculated.

Figure 6 is cited in line 189 before figures 4 and 5 appear in the text.

Lines 239-241 need rewording.

Section 3.3 is more like a discussion.
* * *

---

## Short Comment (SC1) · 26 Sep 2017

In this study, the authors explored seasonal variations of barrier layer thickness (BLT) and seasonal mixed layer heat balance in the Bay of Bengal (BOB). They revealed the importance influence of BLT and vertical mixing on the mixed-layer heat balance in the BOB. Their results are very interesting, and can help to understand the upper-ocean dynamics and air–sea interactions over the BOB. The text is generally well written and the figures are clear. Two minor comments that need to be addressed as listed below.

(1) The Abstract is not concise and clear enough. The new findings of this study should be clearly described in the Abstract.

[Figure]

(2) Line 203, Page 7: "……illustrate the seasonal variability of sub surface conditions……" Replace "sub surface" with "subsurface".

---

## Short Comment (SC2) · 1 Oct 2017

This study examined the seasonal variations of the mixed layer heat balance in the Bay of Bengal, which are mainly associated with the processes of the surface heat flux into the MLD and the vertical mixing at the base of the MLD. The importance of the vertical mixing and the influence of the barrier layer thickness on the mixed-layer heat balance are emphasized in the analysis and the results are generally interesting. Here are some suggestions for the authors to consider in the further revision.

1)The abstract should be shorten and clearly state what's new in this study associated with the previous ones. 2)The "Results" section includes many discussions which often

distract the readers from the results. I would suggest organize those discusions into a "Discussion" section and also revise the "Summary" section to seperate the previous results from other research and the new findings in the current study. 3)Line 135, why choosing a constatnt e folding depth of 25m? 4)Lines 145-150, "ðİŚĎ0 is the surface heat flux adjusted for the penetrative shortwave radiation through the base of the mixed layer...". Should ðİŚĎ0 be the Qnet defined in equation (2)? 5)The text should be go through to avoid some obvious typos and mistakes. For example, Line64, "thesurface" should be "the surface"

---

## Author Comment (AC1) · 6 Oct 2017

**Response to RC1 (25$^{th}$ September 2017)**

**Title: Importance of vertical mixing and barrier layer variation on seasonal mixed layer heat balance in the Bay of Bengal**

We would like to thank referee for the time and effort used to review our manuscript. Your helpful and constructive comments are highly appreciated. This reply addresses all the points highlighted by you.

**Specific comments**

It is useful to state clearly what is new in the study and what has been concluded from the previous in both the Abstract and Summary.

Time series measurements from the Research Moored Array for African-Asian-Australian Monsoon Analysis and Prediction (RAMA) moorings at 15° N, 90° E; 12° N, 90° E; 8° N, 90° E; 4° N, 90° E; 1.5° N, 90° E; 0° N, 90° E are used to investigate the seasonal mixed-layer heat balance and the importance of barrier layer thickness (BLT) and vertical mixing ($Q_{-h}$) in the Bay of Bengal (BoB). It is found that the BLT, $Q_{-h}$ and mixed-layer heat balance all have a strong seasonality in the central BoB. Sea surface temperature (SST), salinity and wind are important for the observed strongest seasonal cycle of BLT in the central BoB, and wind is more important than the SST in the southern BoB. The heat storage rate (HSR) is primarily driven by latent heat flux and shortwave radiation ($Q_{SW}$ $and$ $Q_L$). Seasonal variations and the magnitudes of longwave radiation ($Q_{LW}$), sensible heat flux ($Q_S$), and horizontal mixed-layer heat advection are much weaker compared to those of $Q_{SW}$ $and$ $Q_L$. $Q_{-h}$ follows a pronounced seasonal cycle in the central BoB and is significantly positively correlated with the seasonal cycle of BLT at each mooring location. The seasonal variability of the stability favors the $Q_{-h}$ during winter and summer monsoon and suppress $Q_{-h}$ during monsoon transition periods. We found that $Q_{-h}$ plays the secondary role in the seasonal mixed-layer heat balance in the BoB. It is evident from the analysis that $Q_{-h}$ associated with temperature inversion ($\Delta T$) warms the mixed layer during winter and cools the mixed layer during summer. The warming tendency during winter is strong in the central BoB and weakens towards the equator, indicating a cooling tendency around the year. Our analysis further indicates the weakening of $Q_{-h}$ during monsoon transition periods favors the existence of warmer SST in the BoB, associated with thermal and salinity stratification in the central BoB.

**The following changes were made to the manuscript;**

Time series measurements from the Research Moored Array for African-Asian-Australian Monsoon Analysis and Prediction (RAMA) moorings at 15° N, 90° E; 12° N, 90° E; 8° N, 90° E; 4° N, 90° E; 1.5° N, 90° E; 0° N, 90° E are used to investigate the seasonal mixed-layer heat balance and the importance of barrier layer thickness (BLT) and vertical mixing ($Q_{-h}$) in the Bay of Bengal (BoB). It is found that the BLT, $Q_{-h}$ and mixed-layer heat balance all have a strong seasonality in the central BoB. Sea surface temperature, salinity and wind are important for the observed strongest seasonal cycle of BLT in the central BoB. Consistent with earlier studies, the seasonal mixed-layer heat balance is primarily controls by latent heat flux and shortwave radiation ($Q_{SW} \ and \ Q_L$) and we found that $Q_{-h}$ plays the secondary role compared to the weaker horizontal mixed-layer heat advection in the BoB. It is noted that $Q_{-h}$ is significantly positively correlated with the seasonal cycle of BLT at each mooring location. The seasonal variability of the stability favors the $Q_{-h}$ during winter (high BLT) and summer (relatively low BLT) monsoon and suppress $Q_{-h}$ during monsoon transition periods (Moderate BLT). It is evident from the analysis that $Q_{-h}$ associated with temperature inversion ($\Delta T$) warms the mixed layer during winter. Thermal and salinity stratification is obvious during monsoon transition periods and favors the existence of warmer SST in the BoB due to weakening of $Q_{-h}$. Our analysis further indicates entrainment (E) is more important in $Q_{-h}$ and pointed out the weakening of E associated with BLT tends to weaken the SST-cooling during post-summer monsoon.

I would suggest separating the "Summary and conclusion" section into "Summary" and "Discussion", which will make the reader easy to follow the new findings.

[revised manuscript text omitted]

**Summary:** Consistent with earlier studies, our results reveals that the seasonal mixed-layer heat balance is primarily controls by $Q_{SW}$ and $Q_L$. $Q_{-h}$ plays the seconadary role in mixed-layer heat balance in the BoB. Seasonal variability of BLT influences the $Q_{-h}$ and brings the relative importance to the mixed-layer heat balance in the BoB. Entrainment is more important in $Q_{-h}$ compared to that of vertical diffusion. Sengupta et al. (2008) pointed out that SST-cooling along the track of pre-summer monsoon tropical cyclones (TCs) is about 3 °C, whereas cooling due to post-summer monsoon TCs is ~1 °C. It is evident from our results that BoB is more stable during monsoon transition periods and variability observed in $Q_{-h}$ points out that upper-ocean is more stable during post-summer monsoon. Further it suggests salinity stratification provides more stability compared to thermal stratification in the BoB. The results of this study thus indicate the importance of BLT and vertical mixing on the seasonal mixed-layer heat balance in the BoB. Late phase of summer monsoon and post-summer monsoon are a period of active air-sea interaction, and it is possible that weakening of vertical mixing and strong stratification (higher stability) during this period influence the

intensity and frequency of BoB cyclones. Moreover, studies with systematic measurements are needed to understand the upper-ocean dynamics, the process of vertical mixing and its influence on mixed-layer temperature in the BoB, which can influence the weather and climate in the region and beyond.

Figure 3: it is useful to show the monthly wind speed in Figure 3 at the selected sites in order to make the statement that the wind speed is the main driver of the barrier layer thickness variation in line 192, or make the statement to the discussion.

Thus, it indicates that SST, salinity and wind are important for the observed strongest seasonal cycle of BLT in the central BoB, and that wind is more important than SST for the variability of BLT in the southern BoB (Girishkumar et al., 2011; Felton et al., 2014).

**The following changes were made to the manuscript;**

Thus, it indicates that SST, salinity and wind are important for the observed strongest seasonal cycle of BLT in the central BoB (Girishkumar et al., 2011; Felton et al., 2014).

- The statement is moved to the discussion.

The paragraph starting from line 180: it is not clear how the error bars are calculated.

The estimated mean errors in MLD and ILD are typically ±2 m and ±3 m with a standard deviation of ±8 m and ±18 m, giving errors in BLT of around ±5 m with a standard deviation of ±19 m. MLD in the central BoB exhibits a prominent seasonal variation (Figure 6d) with surface freshening and wind forcing, reaching a maximum in July (42±8 m) when the wind is at its maximum (Babu et al., 2004). ILD varies out of phase with SST, reaching its maximum (87±18 m) in February when SST is minimum and reaching its minimum (16±18 m) in April when SST is at its maximum (Figure 6). The seasonal cycle of BLT varies with ILD, reaching its maximum (69±19 m) in February (highest ILD) and its minimum (2±19 m) in April (lowest ILD) (Figure 3b) then with MLD during summer.

**The following changes were made to the manuscript;**

The estimated standard errors ($SE = standard\ deviation/\sqrt{n}$) in MLD and ILD are typically ±2 m and ±3 m with a standard deviation of ±8 m and ±18 m, giving errors in BLT of around ±5 m with a standard deviation of ±19 m. MLD in the central BoB exhibits a prominent seasonal variation (Figure 6d) with surface freshening and wind forcing, reaching a maximum in July (42±8 m) when the wind is at its maximum (Babu et al., 2004). ILD varies out of phase with SST, reaching its maximum (87±18 m) in February when SST is minimum and reaching its minimum (16±18 m) in April when SST is at its maximum (Figure 6). The seasonal cycle of BLT varies with ILD, reaching its maximum

(69±19 m) in February (highest ILD) and its minimum (2±19 m) in April (lowest ILD) (Figure 3b) then with MLD during summer.

Figure 6 is cited in line 189 before figures 4 and 5 appear in the text.

ILD varies out of phase with SST, reaching its maximum (87±18 m) in February when SST is minimum and reaching its minimum (16±18 m) in April when SST is at its maximum (Figure 6).

**The following changes were made to the manuscript;**

ILD varies out of phase with SST, reaching its maximum (87±18 m) in February when SST is minimum and reaching its minimum (16±18 m) in April when SST is at its maximum.

- Figure 6 is clearly described in section 3.2 mixed-layer heat balance.

Though HSR is primarily driven by $Q_{SW}$ and $Q_L$, the difference observed during summer and winter monsoons brings the importance played by other terms in mixed-layer heat balance as secondary.

**The following changes were made to the manuscript;**

Though HSR is primarily driven by $Q_{SW}$ and $Q_L$, the variations observed during summer and winter monsoons in HSR pointed out the importance of the secondary role played by other terms in the mixed-layer heat balance compared to that of $Q_{SW}$ and $Q_L$.

Section 3.3 is more like a discussion.

- In section 3.3 we compare our new results observed during post-summer monsoon period with previous findings to highlight the role of vertical mixing during that period.

Our results illustrate the shallowest MLD (Figure 6d), associated with a moderate BL (Figure 8), forms during this season in the central BoB. Thus, stratification helps to maintain a warm surface layer, inhibiting SST cooling, and contribution from vertical mixing remains minimum during this period (Figures 8a, 9b). Chacko et al. (2012) suggested the significance of vertical mixing over surface forcing in inducing mixed-layer temperature variability in the BoB and Sengupta et al. (2008) pointed out that during post-summer monsoon the northern BoB responds quite differently to cyclones than during pre-summer monsoon. Stability during post-summer monsoon due to salinity stratification is

stronger compared to that of pre-summer monsoon due to thermal stratification and that may be the reason for the observed differences during cyclone events. Observations from moored RAMA buoys revealed that the importance of seasonal vertical process in SST cooling/warming associated with BLT in the BoB.

**The following changes were made to the manuscript;**

Our results illustrate the shallowest MLD (Figure 6d) associated with a moderate BL (Figure 8) forms during this season in the central BoB. The stratification (relatively higher stability) helps to maintain a warm surface layer, inhibiting SST cooling. Contribution from vertical mixing remains minimum during this period (Figures 8a, 9b). Chacko et al. (2012) suggested the significance of vertical mixing over surface forcing in inducing mixed-layer temperature variability in the BoB and Sengupta et al. (2008) pointed out that during post-summer monsoon the northern BoB responds quite differently to cyclones compared to pre-summer monsoon. Stability during post-summer monsoon because of salinity stratification is stronger compared to that of pre-summer monsoon due to thermal stratification (Figure 5). Thus observations from moored RAMA buoys revealed that the relatively strong stratification and weaker mixing favors the existence of warmer surface layer during post-summer monsoon, and suggests the importance of seasonal vertical mixing in SST cooling/warming associated with BLT in the BoB.

---

## Author Comment (AC2) · 6 Oct 2017

**Response to SC1 (26th September 2017)**

**Title: Importance of vertical mixing and barrier layer variation on seasonal mixed layer heat balance in the Bay of Bengal**

We would like to thank you for the time and effort used to review our manuscript. Your helpful and constructive comments are highly appreciated. This reply addresses all the points highlighted by you.

**Specific comments**

**The Abstract is not concise and clear enough. The new findings of this study should be clearly described in the Abstract.**

Time series measurements from the Research Moored Array for African-Asian-Australian Monsoon Analysis and Prediction (RAMA) moorings at 15° N, 90° E; 12° N, 90° E; 8° N, 90° E; 4° N, 90° E; 1.5° N, 90° E; 0° N, 90° E are used to investigate the seasonal mixed-layer heat balance and the importance of barrier layer thickness (BLT) and vertical mixing ($Q_{-h}$) in the Bay of Bengal (BoB). It is found that the BLT, $Q_{-h}$ and mixed-layer heat balance all have a strong seasonality in the central BoB. Sea surface temperature (SST), salinity and wind are important for the observed strongest seasonal cycle of BLT in the central BoB, and wind is more important than the SST in the southern BoB. The heat storage rate (HSR) is primarily driven by latent heat flux and shortwave radiation ($Q_{SW}$ $and$ $Q_L$). Seasonal variations and the magnitudes of longwave radiation ($Q_{LW}$), sensible heat flux ($Q_S$), and horizontal mixed-layer heat advection are much weaker compared to those of $Q_{SW}$ $and$ $Q_L$. $Q_{-h}$ follows a pronounced seasonal cycle in the central BoB and is significantly positively correlated with the seasonal cycle of BLT at each mooring location. The seasonal variability of the stability favors the $Q_{-h}$ during winter and summer monsoon and suppress $Q_{-h}$ during monsoon transition periods. We found that $Q_{-h}$ plays the secondary role in the seasonal mixed-layer heat balance in the BoB. It is evident from the analysis that $Q_{-h}$ associated with temperature inversion ($\Delta T$) warms the mixed layer during winter and cools the mixed layer during summer. The warming tendency during winter is strong in the central BoB and weakens towards the equator, indicating a cooling tendency around the year. Our analysis further indicates the weakening of $Q_{-h}$ during monsoon transition periods favors the existence of warmer SST in the BoB, associated with thermal and salinity stratification in the central BoB.

**The following changes were made to the manuscript;**

Time series measurements from the Research Moored Array for African-Asian-Australian Monsoon Analysis and Prediction (RAMA) moorings at 15° N, 90° E; 12° N, 90° E; 8° N, 90° E; 4° N, 90° E; 1.5° N, 90° E; 0° N, 90°

E are used to investigate the seasonal mixed-layer heat balance and the importance of barrier layer thickness (BLT) and vertical mixing ($Q_{-h}$) in the Bay of Bengal (BoB). It is found that the BLT, $Q_{-h}$ and mixed-layer heat balance all have a strong seasonality in the central BoB. Sea surface temperature, salinity and wind are important for the observed strongest seasonal cycle of BLT in the central BoB. Consistent with earlier studies, the seasonal mixed-layer heat balance is primarily controls by latent heat flux and shortwave radiation ($Q_{SW}$ $and$ $Q_L$) and we found that $Q_{-h}$ plays the secondary role compared to the weaker horizontal mixed-layer heat advection in the BoB. It is noted that $Q_{-h}$ is significantly positively correlated with the seasonal cycle of BLT at each mooring location. The seasonal variability of the stability favors the $Q_{-h}$ during winter (high BLT) and summer (relatively low BLT) monsoon and suppress $Q_{-h}$ during monsoon transition periods (Moderate BLT). It is evident from the analysis that $Q_{-h}$ associated with temperature inversion ($\Delta T$) warms the mixed layer during winter. Thermal and salinity stratification is obvious during monsoon transition periods and favors the existence of warmer SST in the BoB due to weakening of $Q_{-h}$. Our analysis further indicates entrainment ($E$) is more important in $Q_{-h}$ and pointed out the weakening of $E$ associated with BLT tends to weaken the SST-cooling during post-summer monsoon.

Line 203, Page 7: ". . .. . .illustrate the seasonal variability of sub surface conditions. .. . ." Replace "sub surface" with "subsurface".

The computed vertical temperature gradient (*dT/dz*) (Figure 5a) and salinity gradient (*dS/dz*) (Figure 5b) illustrate the seasonal variability of sub surface conditions.

**The following changes were made to the manuscript;**

The computed vertical temperature gradient (*dT/dz*) (Figure 5a) and salinity gradient (*dS/dz*) (Figure 5b) illustrate the seasonal variability of subsurface conditions.

---

## Author Comment (AC3) · 6 Oct 2017

**Response to SC2 (01$^{st}$ October 2017)**

**Title: Importance of vertical mixing and barrier layer variation on seasonal mixed layer heat balance in the Bay of Bengal**

We would like to thank you for the time and effort used to review our manuscript. Your helpful and constructive comments are highly appreciated. This reply addresses all the points highlighted by you.

**Specific comments**

The abstract should be shorten and clearly state what's new in this study associated with the previous ones..

Time series measurements from the Research Moored Array for African-Asian-Australian Monsoon Analysis and Prediction (RAMA) moorings at 15° N, 90° E; 12° N, 90° E; 8° N, 90° E; 4° N, 90° E; 1.5° N, 90° E; 0° N, 90° E are used to investigate the seasonal mixed-layer heat balance and the importance of barrier layer thickness (BLT) and vertical mixing ($Q_{-h}$) in the Bay of Bengal (BoB). It is found that the BLT, $Q_{-h}$ and mixed-layer heat balance all have a strong seasonality in the central BoB. Sea surface temperature (SST), salinity and wind are important for the observed strongest seasonal cycle of BLT in the central BoB, and wind is more important than the SST in the southern BoB. The heat storage rate (HSR) is primarily driven by latent heat flux and shortwave radiation ($Q_{SW}$ $and$ $Q_L$). Seasonal variations and the magnitudes of longwave radiation ($Q_{LW}$), sensible heat flux ($Q_S$), and horizontal mixed-layer heat advection are much weaker compared to those of $Q_{SW}$ $and$ $Q_L$. $Q_{-h}$ follows a pronounced seasonal cycle in the central BoB and is significantly positively correlated with the seasonal cycle of BLT at each mooring location. The seasonal variability of the stability favors the $Q_{-h}$ during winter and summer monsoon and suppress $Q_{-h}$ during monsoon transition periods. We found that $Q_{-h}$ plays the secondary role in the seasonal mixed-layer heat balance in the BoB. It is evident from the analysis that $Q_{-h}$ associated with temperature inversion ($\Delta T$) warms the mixed layer during winter and cools the mixed layer during summer. The warming tendency during winter is strong in the central BoB and weakens towards the equator, indicating a cooling tendency around the year. Our analysis further indicates the weakening of $Q_{-h}$ during monsoon transition periods favors the existence of warmer SST in the BoB, associated with thermal and salinity stratification in the central BoB.

**The following changes were made to the manuscript;**

Time series measurements from the Research Moored Array for African-Asian-Australian Monsoon Analysis and Prediction (RAMA) moorings at 15° N, 90° E; 12° N, 90° E; 8° N, 90° E; 4° N, 90° E; 1.5° N, 90° E; 0° N, 90° E are used to investigate the seasonal mixed-layer heat balance and the importance of barrier layer thickness (BLT) and vertical mixing ($Q_{-h}$) in the Bay of Bengal (BoB). It is found that the BLT, $Q_{-h}$ and mixed-layer heat balance all have a strong seasonality in the central BoB. Sea surface temperature, salinity and wind are important for the observed strongest seasonal cycle of BLT in the central BoB. Consistent with earlier studies, the seasonal mixed-layer heat balance is primarily controls by latent heat flux and shortwave radiation ($Q_{SW}$ and $Q_L$) and we found that $Q_{-h}$ plays the secondary role compared to the weaker horizontal mixed-layer heat advection in the BoB. It is noted that $Q_{-h}$ is significantly positively correlated with the seasonal cycle of BLT at each mooring location. The seasonal variability of the stability favors the $Q_{-h}$ during winter (high BLT) and summer (relatively low BLT) monsoon and suppress $Q_{-h}$ during monsoon transition periods (Moderate BLT). It is evident from the analysis that $Q_{-h}$ associated with temperature inversion ($\Delta T$) warms the mixed layer during winter. Thermal and salinity stratification is obvious during monsoon transition periods and favors the existence of warmer SST in the BoB due to weakening of $Q_{-h}$. Our analysis further indicates entrainment (E) is more important in $Q_{-h}$ and pointed out the weakening of E associated with BLT tends to weaken the SST-cooling during post-summer monsoon.

The "Results" section includes many discussions which often distract the readers from the results. I would suggest organize those discussions into a "Discussion" section and also revise the "Summary" section to separate the previous results from other research and the new findings in the current study..

[revised manuscript text omitted]

**Summary:** Consistent with earlier studies, our results reveals that the seasonal mixed-layer heat balance is primarily controls by $Q_{SW}$ and $Q_L$. $Q_{-h}$ plays the seconadary role in mixed-layer heat balance in the BoB. Seasonal variability of BLT influences the $Q_{-h}$ and brings the relative importance to the mixed-layer heat balance in the BoB. Entrainment is more important in $Q_{-h}$ compared to that of vertical diffusion. Sengupta et al. (2008) pointed out that SST-cooling along the track of pre-summer monsoon tropical cyclones (TCs) is about 3 °C, whereas cooling due to post-summer monsoon TCs is ~1 °C. It is evident from our results that BoB is more stable during monsoon transition periods and variability observed in $Q_{-h}$ points out that upper-ocean is more stable during post-summer monsoon. Further it suggests salinity stratification provides more stability compared to thermal stratification in the BoB. The results of this study thus indicate the importance of BLT and vertical mixing on the seasonal mixed-layer heat balance in the

BoB. Late phase of summer monsoon and post-summer monsoon are a period of active air-sea interaction, and it is possible that weakening of vertical mixing and strong stratification (higher stability) during this period influence the intensity and frequency of BoB cyclones. Moreover, studies with systematic measurements are needed to understand the upper-ocean dynamics, the process of vertical mixing and its influence on mixed-layer temperature in the BoB, which can influence the weather and climate in the region and beyond.

**Line 135, why choosing a constant e folding depth of 25m?**

The penetrating shortwave radiation below the mixed layer is estimated using $Q_{pen} = 0.47 \times Q_{sw} \cdot e^{-kh}$ (Jouanno et al., 2011), considering a constant $e^-$ folding depth of 25 m ($k=0.04$) and h (MLD).

- Here we use the equation described in Jouanno et al., 2011 to estimate the penetrative shortwave radiation below the mixed-layer. Based on their calculations, we assumed that the shortwave penetration depth is ~25m at all the mooring locations.
- Thangaprakash et al., 2016 (cited in this paper), have estimated the $Q_{pen}$ using, ($Q_{pen} = Q_{shortwave} \times (1 - \alpha)^{-h/\varsigma}$). They have calculated the attenuation depth ($\varsigma$) using MODIS chlorophyll data and pointed out during the study period average attenuation depth is ~20m.

**Lines 145-150, "$Q_0$ is the surface heat flux adjusted for the penetrative shortwave radiation through the base of the mixed layer...".Should $Q_0$ be the $Q_{net}$ defined in equation (2)?**

$$Q_0 = Q_{net} - Q_{pen}$$

$Q_0$ is the surface heat flux adjusted for the penetrative shortwave radiation through the base of the mixed layer.

**The text should be go through to avoid some obvious typos and mistakes. For example, Line64, "thesurface" should be "the surface"**

precipitation and river runoff which exceed evaporation (Harenduprakash and Mitra, 1988), which makes thesurface

**The following changes were made to the manuscript;**

precipitation and river runoff which exceed evaporation (Harenduprakash and Mitra, 1988), which makes the surface

---

## Referee Comment (RC2) · Anonymous Referee #2 · 12 Oct 2017

The authors used time series data from six RAMA moorings to investigate the importance of vertical mixing and the barrier layer (BL) on the ML heat balance in the Bay of Bengal. Study on the vertical mixing, BL and ML heat balance is always interesting for a region like Bay of Bengal (BoB). The figures are of good quality. However, the manuscript needs a major revision before it gets accepted. Below are my concerns for the manuscript.

1. There are few already published works (Girishkumar et al., 2011; Girishkumar et al., 2013) used the RAMA data at the same locations as this study and discussed about the mixed layer heat budget, mechanism of BLT variation, importance of BLT, temperature

[Figure]

inversions and vertical processes on the ML heat budget. So, the authors need to be very specific what is new in this study that was not known from the previous studies.

2. In my opinion, "vertical mixing" is not the right representation of q{-h} in the title and abstract. The term q{-h} in the MLD heat budget equation represents the heat flux through the base of the ML. This represent the vertical mixing at the base of the mixed layer. The way "vertical mixing" used in the title and abstract, it points towards vertical mixing processes in the water column. That's why, when q{-h} is defined as vertical mixing, the statement in line 38 is misleading as the BLT always suppresses vertical mixing. Re-wording "vertical mixing" in the title and the abstract would be useful.

3. The study used data from the RAMA moorings at location 15, 12, 8, 4, 1.5 and 0 N. Then why the figures 3-7 and 9 do not show fields at all the RAMA locations? It is very hard to follow the results comparing the central and southern BoB.

4. A figure or table for the RAMA data coverage will be useful.

5. What is the justification for selecting the averaging months for summer monsoon? In general May is considered as the pre-monsoon because the summer monsoon sets in around the beginning of June.

6. Figure 4: At which RAMA location?

7. Figure 5: Why there is a patch of the higher amplitude of temperature gradient and stability at 40m depth?

8. The poor vertical resolution of the data raises concern about how well the stratification has been resolved from this data. The authors can check the stability profiles computed from nearby other observation data with higher vertical resolution.

9. It is not clear why NCOM fields were used. How accurate are the NCOM fields in this region? 10. Line 118: The MLD criterion is not clear. Is it density change by 0.125kg/mˆ3 or density change equivalent to 0.8C temperature change?

11. Line 165: Section 3.1 discusses about ILD, MLD, BLT, stratification which are not the surface conditions. Then why "surface conditions" in the section title?

12. Line 221-223: Where is the evidence? Any figure or reference?

13. Separating the importance of q{-h} and BLT into two subsections might be useful.

14. Line 325: What is the "missing source"?

15. Table 1: Why correlation is smallest at 4N and higher towards north and south?

---

## Author Comment (AC4) · 18 Oct 2017

**Response to RC2 (12th October 2017)**

**Title: Importance of vertical mixing and barrier layer variation on seasonal mixed layer heat balance in the Bay of Bengal**

We would like to thank the referee for the time and effort used to review our manuscript. Your helpful and constructive comments are highly appreciated. This reply addresses all the points highlighted by you.

**Specific comments**

There are few already published works (Girishkumar et al., 2011; Girishkumar et al., 2013) used the RAMA data at the same locations as this study and discussed about the mixed layer heat budget, mechanism of BLT variation, importance of BLT, temperature inversions and vertical processes on the ML heat budget. So, the authors need to be very specific what is new in this study that was not known from the previous studies.

01. Girishkumar et al., 2011 use data from RAMA mooring at 8°N, 90°E with remote sensing data to discuss the intra-seasonal variability of BLT in the south central BoB from November 2006 to April 2009. They points out that the observed intra-seasonal variability during the study period is mainly due to the movement of ILD in the presence of shallow mixed-layer. Further they suggests that both ILD and BLT are modulated by the westward propagating intra-seasonal Rossby waves.

02. Girishkumar et al., 2013 use data from RAMA mooring at 8°N, 90°E for two winters (2006-07, 2007-08) to pointed out the importance of temperature inversions during the winter season. They discuss the influence of temperature inversions and BLT on mixed-layer heat budget. Further they suggests temperature inversions are associated with thicker BL and heating by penetrative solar radiation below the mixed-layer is greater than the heating of the mixed-layer by net surface heat flux and horizontal advection.

03. In this study we use observation data from six RAMA moorings located along 90°E from January 2008 to December 2016 with CTD, satellite and model data to study the importance of vertical mixing and BL variation on seasonal mixed-layer heat budget in the BoB. We points out that the seasonal cycle of BLT, vertical mixing and mixed-layer heat storage are prominent at 15°N, 90°E and weakens towards the equator. Further we points out the positive correlation between BLT and vertical mixing at all mooring locations. Then we discuss about the seasonal stability at the mooring locations and points out the importance of monsoon transition periods. The BoB is more stable during monsoon transition periods due to thermal and salinity stratification and the vertical mixing remains relatively small during this period. Then we points out that the vertical mixing plays the secondary role in the mixed-layer heat balance. Further we suggests that the entrainment is more important in the vertical mixing process in the BoB based on our results. Strong salinity stratification and moderate BLT thickness reduce the vertical mixing during post-summer monsoon and helps to maintain warmer SSTs in the BoB. We use NCOM and CTD profile data to provide evidences for the conditions estimated at RAMA mooring locations. Further based on the strongest seasonal cycles observed and data availability, we select mooring at 15°N, 90°E to explain the conditions in the central BoB in detail.

    The results of this study thus discuss the importance of BLT and vertical mixing on the seasonal mixed-layer heat balance in the BoB. Further it brings the importance of variability of these conditions during post-summer monsoon, a crucial period for active air-sea interaction in this region.

In my opinion, "vertical mixing" is not the right representation of q{-h} in the title and abstract. The term q{-h} in the MLD heat budget equation represents the heat flux through the base of the ML. This represent the vertical mixing at the base of the mixed layer. The way "vertical mixing" used in the title and abstract, it points towards vertical mixing processes in the water column. That's why, when q{-h} is defined as vertical mixing, the statement in line 38 is misleading as the BLT always suppresses vertical mixing. Re-wording "vertical mixing" in the title and the abstract would be useful.

- We have replaced the word "vertical mixing" with "vertical heat flux"

The study used data from the RAMA moorings at location 15, 12, 8, 4, 1.5 and 0 N. Then why the figures 3-7 and 9 do not show fields at all the RAMA locations? It is very hard to follow the results comparing the central and southern BoB.

[Figure]

- We have edited the figure 3 as it explain the conditions at all the mooring locations selected in this study. New figure illustrates the variations of subsurface temperature, salinity, MLD, ILD, BLT and 23°C isothermal layer at the mooring locations. Then we selected the mooring at 15°N, 90°E to indicate the seasonal variability in detail due to its data availability and strong seasonal cycles. The figures 4-7 explains the seasonal conditions at central BoB.
- Figure 8 clearly illustrates the variability of vertical process and BLT from central BoB to equator. As we are discussing about the importance of vertical process and BLT on the mixed-layer heat balance, we use figure 8 to comparatively explain the changes in vertical process and BLT at all the mooring locations.
- Finally we select the mooring location with the highest variability observed in BLT and vertical process to explain the conditions in detail. So figure 9 illustrates the importance of entrainment in mixed layer heat balance at 15°N, 90°E.

**The following changes were made to the manuscript;**

Line 180 to 183,

Next, we examine the stratification in the upper 140 m at the RAMA locations in the BoB. The moorings are located from the central BoB to the equator, which are in a region of strong seasonal variability of BLT (Figure 1). We have selected the mooring at 15°N, 90°E (Figures 3a, 3b), which has the longest

Next, we examine the stratification in the upper 120 m at the RAMA locations in the BoB. The moorings are located from the central BoB to the equator, which are in a region of strong seasonal variability of BLT (Figure 1). Figure 4 shows the variability of MLD, ILD, BLT and 23°C isotherm (D23) with subsurface temperature and salinity at each RAMA mooring location. We have selected the mooring at 15°N, 90°E (Figures 4a, 4b), which has the longest

[Figure]

**Figure 4.** Time series of daily averaged RAMA data from January 2008 – December 2016 at 15°N, 12°N, 8°N, 4°N, 1.5°N, and 0°N, 90°E. (a, b, c, g, h, i) subsurface temperature measured by RAMA moorings, (d, e, f, j, k, l) subsurface salinity measured by RAMA buoy moorings. In the figure: thick line (MLD), thin line (ILD), thick-dashed line (D23) and thin-dashed line (BLT).

A figure or table for the RAMA data coverage will be useful.

**The following changes were made to the manuscript;**

[Figure]

**Figure 3.** Availability of daily measurements from January 2008 to December 2016 at RAMA moorings in the BoB. Blue color represents temperature (T), red color represents salinity (S), green color represents shortwave radiation ($Q_{sw}$) and purple represents currents (u, v) measured at 10 m depth (Cur).

- Figure explains the availability of temperature (T) and salinity (S) up to 120m, shortwave radiation (Qsw) and currents at 10m depth (Cur) at all the mooring locations from January 2008 to December 2016.

What is the justification for selecting the averaging months for summer monsoon? In general May is considered as the pre-monsoon because the summer monsoon sets in around the beginning of June.

- When selecting the averaging months for summer monsoon authors have used either May or June as the onset month. For example: May to September (Shankar et al., 2002). June to September (Rao et al., 2012). Some have been very specific, 6th May to 24th September (Warner et al., 2016).
- In this study, based on average wind observations by RAMA moorings at 15°N, 12°N and 8°N 90°E for the 9 years we consider the mid-May as the onset of the summer monsoon. Then we select mid-May to September as the months for summer monsoon.

**The following changes were made to the manuscript;**

BL almost disappears during pre-summer monsoon (March-April) associated with low precipitation (Figure 1b), low surface freshening and warmer SST (Figure 2b). During summer monsoon (May-September), BLT is relatively thicker in the eastern boundary of the BoB, associated with higher precipitation (Figure 1c) and surface freshening (Figure 2c).

BL almost disappears during pre-summer monsoon (March – mid-May) associated with low precipitation (Figure 1b), low surface freshening and warmer SST (Figure 2b). During summer monsoon (mid-May –September), BLT is relatively thicker in the eastern boundary of the BoB, associated with higher precipitation (Figure 1c) and surface freshening (Figure 2c).

**Figure 4: At which RAMA location?**

**Figure 4.** Comparison of NCOM (red) and RAMA (blue) estimated monthly climatologies of (a) MLD, (b) ILD, and (c) BLT.

**The following changes were made to the manuscript;**

**Figure 4.** Comparison of NCOM (red) and RAMA (blue) estimated monthly climatologies of (a) MLD, (b) ILD, and (c) BLT at 15°N, 90°E.

**Figure 5: Why there is a patch of the higher amplitude of temperature gradient and stability at 40m depth?**

- The patch of the higher amplitude of temperature gradient and stability at 40m depth is due to an error in the calculation of temperature gradient. The error associated with data at 40m depth was resolved.

[Figure]

[Figure]

**The following changes were made to the manuscript;**

[Figure]

The poor vertical resolution of the data raises concern about how well the stratification has been resolved from this data. The authors can check the stability profiles computed from nearby other observation data with higher vertical resolution.

To address this issue we have used CTD data from National Oceanographic Data Center (NODC) from January 2000 to December 2016 (www.nodc.noaa.gov). We have selected the region covering the RAMA mooring at 15°N, 90°E and there were 68 CTD profiles in the region (14-18°N, 88-92°E) from April 2007 to April 2016. CTD profiles were not available during May and December. The calculated stability from CTD follows a similar pattern with the results from RAMA and NCOM data.

**The following changes were made to the manuscript;**

Line 153 to 155,

Further we use the Navy Coastal Ocean Model (NCOM) monthly climatological data (from 1990 to 2011) (Ke Huang et al., 2015) to compare with the observed seasonal variability in upper layer stratification, subsurface temperature and salinity gradients, and the stability at 15°N, 90°E by the RAMA mooring.

Further we use the Navy Coastal Ocean Model (NCOM) monthly climatological data (from 1990 to 2011) (Ke Huang et al., 2015) and CTD data from National Oceanographic Data Center (NODC) to compare with the observed seasonal variability in upper layer stratification, subsurface temperature and salinity gradients, and the stability at 15°N, 90°E by the RAMA mooring. We have selected the region covering the RAMA mooring at 15°N, 90°E and there were 68 CTD profiles in the region (14-18°N, 88-92°E) from April 2007 to April 2016. CTD profiles were not available during May and December.

Line 205 to 210,

The estimated upper ocean stability illustrates that the upper ocean layers at 15°N, 90°E are more stable during monsoon transition periods compared to that of winter and summer (Figure 5c). Thus the results pointed out that winter and summer favors the vertical mixing (Thangaprakash et al., 2016) with the presence of more unstable layers in the central BoB, and pre and post-summer monsoon tends to inhibit the vertical mixing due to the presence of more stable water layers.

The estimated upper ocean stability illustrates that the upper ocean layers at 15°N, 90°E are more stable during monsoon transition periods compared to that of winter and summer (Figure 6c). It is evident from the estimated mean stability in the region (14-18°N, 88-92°E) using 68 CTD profiles (Figure 6d). Thus the results pointed out that winter and summer favors the vertical mixing (Thangaprakash et al., 2016) with the presence of more unstable layers in the central BoB, and pre and post-summer monsoon tends to inhibit the vertical mixing due to the presence of more stable water layers.

[Figure]

**Figure 6.** Comparison of upper ocean stability estimated from NCOM (contour) and RAMA (color shaded) at 15°N, 90°E. (a) Temperature gradient (positive when temperature decreases downward), (b) salinity gradient (positive when salinity increases downward), (c) stability (in terms of buoyancy frequency), and (d) stability estimated for individual months using 68 CTD profiles (NODC) from April 2007 – April 2016 for the region 14-18°N, 88-92°E. In figure d, gray color (individual profiles), red color (mean), and the numbers in boxes represent number of profiles available for each month.

It is not clear why NCOM fields were used. How accurate are the NCOM fields in this region?

We use NCOM fields to compare with the vertical structure observed at RAMA locations. Also, compared to other model data NCOM has a good vertical resolution. The stability estimated from CTD profiles also provide similar pattern to the results computed using NCOM and RAMA data.

NCOM field have been used in Ke Huang et al., 2015 for the Indian Ocean including BoB.

Line 118: The MLD criterion is not clear. Is it density change by 0.125kg/m^3 or density change equivalent to 0.8C temperature change?

- In this study we have considered the density change equivalent to 0.8°C temperature change from the surface to estimate the MLD.

**The following changes were made to the manuscript;**

MLD is defined as the depth where the density has changed by 0.125 kg m$^{-3}$ ($\Delta T = 0.8\ °C$) from the surface value at 1 m (Rao and Sivakumar, 2000; Kara et al., 2003).

MLD is defined as the depth where the density changed is equivalent to 0.8 $°C$ temperature change from the surface value at 1 m (Rao and Sivakumar, 2000; Kara et al., 2003).

Line 165: Section 3.1 discusses about ILD, MLD, BLT, stratification which are not the surface conditions. Then why "surface conditions" in the section title?

- In this study section 3.1 discusses about climatological conditions in the Bay of Bengal.

**The following changes were made to the manuscript;**

3.1 Variability of climatological surface conditions in the BoB

3.1 Variability of climatological conditions in the BoB

Line 221-223: Where is the evidence? Any figure or reference?

Wind speed undergoes a more pronounced seasonal cycle at 15°N and 12°N compared to that at 8°N, 4°N, 1.5°N and 0°N, tending to enhance the seasonal cycle of $Q_L$ in the central BoB.

- We have concluded this from the observation at RAMA moorings. Generally latent heat loss is affected by the wind. We did not include the figure in the manuscript.
- The figure below clearly illustrates the conditions stated in line 221-223.

[Figure]

Separating the importance of q{-h} and BLT into two subsections might be useful.

**The following changes were made to the manuscript;**

- We have seperated the importance of vertical process and BLT. The results are explained in **Section 3.3**.

Line 325: What is the "missing source"?

We have found a missing source of warming during August–September in the central BoB up to ~25 Wm$^{-2}$. The uncertinities are mainly associated with measurement errors, calculation errors and parameterization of the vertical process.

**The following changes were made to the manuscript;**

We have found a missing source of warming in $Q_{-h}$ (when estimated as the residual term) during August–September in the central BoB up to ~25 Wm$^{-2}$. Estimated entrainment and vertical diffusion remains relatively small and indicates a cooling tendency during that period. The uncertinities may be associated with measurement errors, calculation errors and parameterization of the vertical process.

Table 1: Why correlation is smallest at 4N and higher towards north and south?

[Figure]

According to the figure (figure 8 in the text) the BLT remains relatively constant from mid-December to mid-June at 4°N, 90°E. During the same time period the contribution of $Q_{-h}$ also remains relatively constant. That is the reason for the observed correlation between BLT and $Q_{-h}$ at this location. Similar to observations at other mooring locations the correlation is positive at 4°N, 90°E, but it is not strong due to the almost constant BLT.

**The following changes were made to the manuscript;**

Line 313 to 316,

The horizontal mixed-layer heat advection also weaker compared to that of vertical mixing. The vertical mixing at the base of the mixed layer ($Q_{-h}$), estimated as the residual in the heat balance following Foltz and McPhaden (2009), also follows a pronounced seasonal cycle in the central BoB, and is correlated positively with the seasonal cycle of BLT at each mooring location. We find that

The horizontal mixed-layer heat advection also weaker compared to that of vertical mixing. The vertical mixing at the base of the mixed layer ($Q_{-h}$), estimated as the residual in the heat balance following Foltz and McPhaden (2009), also follows a pronounced seasonal cycle in the central BoB, and is correlated positively with the seasonal cycle of BLT at each mooring location (Table 1). At 4°N, 90°E RAMA mooring the estimated BLT and $Q_{-h}$ remains relatively constant from mid-December to mid-June (Figure 9) and because of that the computed correlation is smallest compared to other locations. We find that